# DNA Satellites Impact Begomovirus Diseases in a Virus-Specific Manner

**DOI:** 10.3390/ijms26125814

**Published:** 2025-06-17

**Authors:** Vincent N. Fondong

**Affiliations:** Department of Biological Sciences, Delaware State University, Dover, DE 19901, USA; vfondong@desu.edu

**Keywords:** alphasatellites, betasatellites, deltasatellites, begomoviruses

## Abstract

Begomoviruses infect many crops and weeds globally, especially in the tropical and subtropical regions, where there are waves of epidemics. These begomovirus epidemics are frequently associated with three DNA satellites: betasatellites, alphasatellites, and deltasatellites. Except for the origin of replication, these satellites show no sequence identity with the helper begomovirus. Alphasatellites and betasatellites encode the α-Rep and βC1 proteins, respectively, while deltasatellites encode no proteins. α-Rep, which functions like the Rep of the helper begomoviruses, ensures alphasatellite replication autonomy, while betasatellites and deltasatellites depend wholly on the helper virus for replication. The betasatellite βC1 protein is a pathogenicity determinant and suppressor of RNA silencing. The associations between satellites and helper viruses vary, depending on the virus and the host, and the roles of these satellites in disease development are an active area of investigation. This review highlights current information on the role of DNA satellites in begomovirus diseases and examines commonalities and differences between and within these satellites under prevailing conditions. Furthermore, two episomes, SEGS-1 and SEGS-2, associated with cassava mosaic geminiviruses, and their possible status as DNA satellites are discussed. DNA satellites are a major factor in begomovirus infections, which are a major constraint to crop production, especially in tropical and subtropical regions. Thus, areas for future research efforts, as well as implications in the biotechnological management of these viruses, are discussed in this review.

## 1. Introduction

Geminiviruses (family *Geminiviridae*) are the largest known group of plant viruses and are of global economic concern as they cause devastating diseases in many important crops. The family *Geminiviridae* comprises fourteen genera—*Begomovirus*, *Capulavirus*, *Citlodavirus*, *Curtovirus*, *Grablovirus*, *Maldovirus*, *Mastrevirus*, *Mulcrilevirus*, *Opunvirus*, *Topilevirus*, *Topocuvirus*, *Turncurtovirus*, *Becurtovirus*, and *Eragrovirus*—based on genome organization, host range, and insect vector [1]. Geminiviruses have circular, single-stranded DNA (ssDNA) genomes, and, except for the genus *Begomovirus*, which has some members with two-component (bipartite) genomes (DNA-A and DNA-B), have single-component (monopartite) genomes. Viral DNA replication follows a combination of rolling circle and recombination-dependent modes of replication, which produce double-stranded (ds) DNA replication form intermediates [2].

Gene transcription in geminiviruses occurs bidirectionally from the dsDNA template from divergent promoters located in a DNA region containing cis-acting signals required for gene transcription and DNA replication [3]. DNA-A components of bipartite begomoviruses contain five or six canonical genes—one or two are in the sense strand and four are in the complementary sense strand. The sense strand genes are *AV1*, which encodes the coat protein (CP), and *AV2*, which is found in Old World bipartite begomoviruses, but not in New World viruses. The four genes in the complementary sense are *AC1*, whose gene product is the replication-associated protein (Rep); *AC2* and *AC3*, which code for the transcription activation protein (TrAP) and the replication enhancer protein (REn); and AC4, which is located within AC1 at the 5’ end but in a different reading frame [4] (Figure 1). DNA-B has two genes, *BV1* and *BC1*, which code for the movement protein (MP) and nuclear shuttle protein (NSP), respectively. DNA-A and DNA-B show no sequence similarities, except for an approximately 200-nucleotide segment in the 5′ intergenic region (IR), designated as the common region (CR). The CR contains an origin of replication organized modularly, including a stem–loop structure containing the invariant nonanucleotide TAATATTAC or TAAGATTCC sequence, where cleavage, synthesis, and joining of the DNA virion strand occur during replication [5,6]. All monopartite geminivirus proteins are encoded in one genomic component. The ORFs include the virion-sense *V1* (CP) and *V2* (MP) genes and the complementary-sense *C1* (Rep), *C2* (TrAP), *C3* (REn), and *C4* (C4) genes. Analogous to bipartite begomoviruses, the IR harbors sequence elements found in the CR of bipartite begomoviruses, including the invariant nonanucleotide TAATATTAC sequence that harbors the origin of replication (Figure 1B).

Although RNA satellites have long been found in association with RNA viruses [7,8], the first report of a DNA satellite was in 1997, when a DNA satellite was found in northern Australia in association with tomato leaf curl virus (ToLCV), a monopartite *begomovirus* [9]. Since then, many begomoviruses have been found in association with three classes of satellite molecules, namely, betasatellites, alphasatellites, and deltasatellites. Like the helper virus, these satellites have a circular ssDNA genome and replicate using a combination of rolling circle and recombination-dependent replication (RDR) from the dsDNA replication intermediate. The RDR mode of geminivirus replication is similar to the replication mechanism of the bacteriophage φ29. Thus, the rolling circle amplification (RCA) approach, which is based on the φ29 DNA polymerase, was developed and has been used extensively to identify previously undetected viruses and DNA satellites. The key advantage of this detection method is that amplification does not require previous knowledge of nucleotide sequences [10,11]. The increase in the number of DNA satellites associated with begomoviruses is, therefore, likely due mainly to the availability of RCA as opposed to the recent emergence of these molecules.

### 1.1. Structure and Function of Begomovirus Satellite Molecules

Alphasatellites and betasatellites are half the size (~1.3 kb) of their helper begomovirus genome (for monopartite begomoviruses) or genome component (for bipartite begomoviruses). Like the helper viruses, these DNA satellites have circular, ssDNA genomes and contain a putative origin of replication marked by a canonical nonanucleotide motif (TAA/GTATTAC) in the loop region of the stem–loop structure. Both DNA satellites encode at least one protein. In contrast, deltasatellites, which are approximately ~700 nucleotides in size, encode no proteins. The specific structural features of each DNA satellite are detailed below.

### 1.2. Betasatellites (Family Tolecusatellitidae)

Betasatellites encode in the complementary strand, a 13.5 kDa protein, known as βC1. βC1 proteins from several betasatellites have been shown to increase levels of viral DNA and pronounced viral disease symptoms by suppressing the host antiviral RNA-silencing properties [12,13,14,15,16,17]. Some betasatellites have been found to encode a second protein in the virus sense of the molecule, designated as βVl, which is also a pathogenicity determinant [18,19]. Betasatellite genomes also contain a conserved sequence known as satellite conserved region (SCR), with an approximate size of ~125 nt (Figure 2A), and a 160-to-280 nt adenine-rich (A-rich) (or AT-rich) region. A search of inverted repeats with the potential to form stem–loop structures with a stem of at least six nucleotides and a loop of at least four nucleotides identified only one such repeat, which harbors the TAATATTAC nonanucleotide sequence containing the origin of replication [20]. Betasatellites have the same origin of virion-strand replication as the helper viruses and depend wholly on the helper virus for replication. Although the SCR was suggested to be the site where the helper begomovirus Rep binds to initiate replication [21,22,23], more recent studies suggest that Rep trans-replication of betasatellites is mediated by sequences between the A-rich region and the SCR [24]. Unlike some Old World monopartite begomoviruses, begomoviruses found in the New World, where betasatellites are absent [25,26,27], have highly specific Rep-binding sites in DNA-A and DNA-B [20].

Although a biological function has not yet been assigned for the betasatellite A-rich region, in alphasatellites, the A-rich region was suggested to have been used to increase the size of the nanovirus DNA component from which alphasatellites are believed to have evolved [21,28]. Thus, by analogy, the betasatellite A-rich region might have the same origins and role.

### 1.3. Alphasatellites (Family Alphasatellitidae)

Alphasatellites are of a similar size to betasatellites, approximately 1.3 kb. The alphasatellites encode the ~37 kDa α-Rep, which functions like the geminivirus Rep protein and ensures autonomous replication [26]. Similarly to betasatellites, alphasatellites have an A-rich region and a predicted hairpin structure (Figure 2B). A-rich sequences are thought to increase the sizes of alphasatellite molecules to half the size of the begomovirus components [29] and might have been acquired from nanoviruses [30,31]. A similar search for potential stem–loop structures as that described for betasatellites identified a stem–loop structure harboring the nonanucleotide TAGTATTAC, common to nanoviruses and analogous to the TAATATTAC nonanucleotide sequence in helper virus stem–loop structures (Figure 2B). The slight difference between the nonanucleotide loop sequences of alphasatellites and helper geminiviruses can, at least partly, explain why alphasatellites self-replicate using α-Rep, which is more specific for the alphasatelilite loop sequence. Alphasatellites have at least three other predicted stem–loop structures (Figure 2B), the biological relevance of which will need to be examined.

### 1.4. Deltasatellites (Family Tolecusatellitidae)

The least studied of the three begomovirus DNA satellites identified so far are deltasatellites, which, unlike betasatellites and alphasatellites, do not code for a protein, and the molecules are half the size of betasatellites and alphasatellites. Three groups of deltasatellites: deltasatellites associated with the monopartite ToLCV in Australia [9], those associated with sweet potato viruses in Spain and Venezuela [28,32], and deltasatellites associated with New World bipartite begomoviruses, including sida golden yellow vein virus (SiGYVV) [28,33,34]. Structurally, deltasatellites associated with ToLCV are similar to the one associated with the New World bipartite begomovirus, SiGYVV, and both are characterized by the SCR, conserved stem–loop, and A-rich region (Figure 2C). The SCR is absent in SPLCV-SPLCD1, the deltasatellite associated with sweet potato leaf curl virus (SPLCV). Half of the SPLCV-SPLCD1 genome has sequence identity to a truncated *C1* and IR of SPLCV and was likely acquired from a recombination event (Figure 2C).

Structurally, stem–loop structures are found to be the most common features in these satellite molecules. In addition to harboring the origin of replication, stem–loop structures provide specificity during virus genome assembly and packaging. In ssRNA viruses, for example, packaging proteins specifically recognize and bind to the stem–loop with high affinity, followed by encapsidation of only its genetic material [35,36,37,38,39,40].

## 2. Role of DNA Satellites in Virus Disease Development

Begomoviruses cause important crop diseases and are a threat to global agriculture. Symptoms displayed by plants infected by these viruses include leaf curling, distortion, yellowing, mosaic patterns, stunting, and vein swelling, often leading to substantial yield losses. Begomoviruses are geographically separated into New World and Old World viruses. Most New World begomoviruses have a bipartite genome, while Old World viruses tend to have monopartite genomes, which are frequently associated with DNA satellites. Alphasatellites are found in the Old World and New World. In the Old World, they are associated with monopartite begomoviruses and, in many cases, coinfect the same plant with betasatellites, while in the New World, they are associated with bipartite begomoviruses [27]. Like alphasatellites, deltasatellites are associated with begomoviruses in the Old World and New World, where betasatellites are absent.

### 2.1. Interaction Between Betasatellites and Associated Helper Begomoviruses Is Synergistic

Begomovirus–betasatellite complexes have been reported in several crops, including cotton (*Gossypium hirsutum*), pepper (*Capsicum* spp.), okra (*Abelmoschus esculentus*), radish (*Raphanus sativus*), papaya (*Carica papaya* L.), and tomato (*Solanum lycopersicum*). In the Old World, betasatellites have been shown to act synergistically in mixed infections with helper viruses, particularly in *Malvaceae* and *Solanaceae* hosts, but, interestingly, not in some *Cucurbitaceae* hosts. In Pakistan and India, synergistic interactions were reported between radish leaf curl virus (RaLCV) and its betasatellite, RaLCB, in infected radish plants [41,42]. Furthermore, studies in China reported a synergistic interaction between tomato yellow leaf curl China virus (TYLCCNV) and tomato yellow leaf curl China betasatellite (TYLCCNB) in tomato [43]. Moreover, pepper chili leaf curl betasatellite (ChLCB) and pepper leaf curl Lahore virus (PepLCLV), which likely resulted from a recombination event between papaya leaf curl virus (PaLCuV) and chili leaf curl virus (ChiLCV), caused more severe symptoms in pepper plants compared with a single infection by PepLCLV [44]. The study further showed that ChiLCV and ChLCB coinfect potato (*Solanum tuberosum*) in nature, thus indicating a broad host range for these betasatellites.

Although the whitefly is the vector of begomoviruses and associated DNA satellites and is, therefore, an important component in begomovirus disease epidemics, most studies on the incidence, distribution, and diversity of these viruses and DNA satellites have disproportionately been on plant hosts. In one of the few studies carried out so far, four cryptic whitefly species were sampled in pumpkin, cucumber, tomato, sponge gourd, and ridge gourd fields in India. Analysis of these samples identified four types of betasatellites—tomato leaf curl Bundi betasatellite (ToLCBDB), okra enation leaf curl betasatellite (OLCuB), Bhendi yellow vein mosaic betasatellite (BYVB), and papaya leaf curl virus betasatellite (PaLCB)—as well as the alphasatellite ageratum enation alphasatellite (AEV) and cotton leaf curl Multan alphasatellite (CLCuMuA) [45]. This stresses the importance of monitoring the incidence, distribution, and diversity of these viruses and associated satellites in whitefly vectors.

Much of sub-Saharan Africa, especially West Africa, has rich vegetation due to uniformly high temperatures and heavy rainfall year-round. This provides a rich environment for plant viruses, which infect both cultivated crops and weeds. Unfortunately, until recently, there has been a dearth of information on the occurrence of begomoviruses and associated DNA satellites in much of this region. A study carried out in the tropical rainforest part of Cameroon identified cotton leaf curl Gezira virus (CLCuGeV) and okra yellow crinkle virus (OYCrV), respectively, in coinfection with okra cotton leaf curl Gezira betasatellite (CLCuGeB) [46]. This study followed a previous study by the same group that reported the coinfection of *Ageratum conyzoides* (family: *Asteraceae*) by a previously unreported ageratum leaf curl Cameroon virus (ALCCMV) and its associated ageratum leaf curl Cameroon betasatellite (ALCCMB) [47]. *A. conyzoides* is an invasive weed that grows extensively in much of West Africa and, therefore, likely acts as a reservoir of these viruses and their DNA satellites. Furthermore, still in the rainforest zone of West Africa, a previously unreported betasatellite associated with ToLCD, designated as tomato leaf curl Togo betasatellite (ToLCTGB), was found to be associated with tomato leaf curl Kumasi virus (ToLCKuV) and tomato leaf curl Nigeria virus (ToLCNGV) [48]. Similarly to the previously reported tobacco curly shoot disease in China [49], ToLCTGB was found not to enhance ToLCD symptoms in tomato. In contrast, it increased the symptoms of both viruses in *Nicotiana* spp. and *Datura stramonium* [48]. The same group found in Mali, in the Sudano-Sahelian climate zone, that tomato yellow leaf curl Mali virus (TYLCMLV), a recombinant virus, and cotton leaf curl Gezira betasatellite (CLCuGB) caused severe symptoms in tomato and several other solanaceous hosts, but not in common bean [50]. Another study in Burkina Faso, which, like Mali, is in the Sudano-Sahelian zone, reported the coinfection of okra by CLCuGB and its helper virus, CLCuGeV [51]. Furthermore, a study conducted in Uganda, in eastern Africa, detected vernonia crinkle virus (VeCrV) and its associated and previously unidentified vernonia crinkle betasatellite (VeCrB) in African bitter leaf (*Vernonia amygdalina*), a non-solanaceous vegetable that grows in tropical Africa [52].

#### βC1 RNA-Silencing Suppression Function Contributes to Betasatellite Enhancement of Helper Virus Symptoms

The ability of βC1 to suppress antiviral RNA silencing is partly due to its interaction with several proteins involved in the RNA-silencing pathway, including S-adenosyl homocysteine hydrolase (SAHH) [43], SnRK1 (sucrose non-fermenting1-related protein kinase 1), calmodulin-like protein (rgs-CaM) [53], and Argonaute 1 (AGO1) [54]. These interactions may inhibit methylation, antiviral phosphorylation, and formation of the RNA-induced silencing complex (RISC), thereby suppressing RNA silencing. This action stabilizes the begomovirus–betasatellite complexes and increases viral DNA levels, as well as symptom severity.

βC1′s ability to antagonize host defenses is not limited to suppression of RNA silencing, given that it is also involved in virus movement. Specifically, it was shown that βC1 interacts with the CP of Bhendi yellow vein mosaic virus (BYVMV), the helper virus, as well as with the host karyopherin α, a nuclear importin-like protein, to facilitate shuttling between the nucleus and the cytoplasm [55]. Furthermore, βC1 forms a complex with Asymmetric leaves 1 (AS1), a gene that facilitates the specification of leaf adaxial identity [56] and attenuates the expression of selective jasmonic acid (JA)-responsive gene, leading to suppression of JA host defense responses [17]. Additionally, βC1 of CLCuMuB associated with cotton leaf curl disease interacts with the ubiquitin-conjugating enzyme (UBC), thereby interfering in the ubiquitin–proteasome pathway, which results in the development of leaf-curling symptoms [57]. Sequence analysis shows that except for a few conserved residues, including the PFDFN motif found in the middle of the protein, βC1 exhibits considerable sequence variation, especially at the proteins’ N- and C-termini (Figure 3A). These studies show the important role played by βC1 in betasatellite induction of severe symptoms during coinfection with helper viruses.

### 2.2. Alphasatellites Tend to Attenuate Symptoms in Coinfections with Helper Viruses

Alphasatellites have been found to not increase disease symptoms or helper viral DNA accumulation. Many alphasatellites tend to modulate virulence and symptom development. For example, a cassava mosaic virus-associated alphasatellite, closely related (80% sequence identity) to cotton leaf curl Gezira (CLCuGeA), was found not to influence cassava mosaic disease (CMD) symptoms in coinfected cassava in Madagascar [58]. The same was observed for the interaction between CLCuGeA and okra leaf curl Burkina Faso alphasatellite (OLCuBFA), and a complex of CLCuGeV and OYCrV in okra plants in Cameroon [46]. Furthermore, two alphasatellites associated with the cotton leaf curl Multan disease in India were reported to attenuate and delay symptom development in *N. benthamiana* [59].

In the Americas, melon chlorotic mosaic virus-associated alphasatellite (MeCMA) was detected in watermelon samples collected from Venezuela in association with its helper virus, melon chlorotic mosaic virus (MeCMV), a bipartite begomovirus [60]. Subsequent studies identified MeCMA in association with several bipartite begomoviruses infecting non-cultivated plants and watermelon crops [61,62]. In an in-depth survey of begomovirus-associated satellite DNAs conducted in the Old World and New World (USA, Guatemala, Israel, Puerto Rico, and Spain), whiteflies were collected in weeds, as well as in bean, eggplant, pumpkin, squash, and tomato fields, and then analyzed using vector-enabled metagenomic (VEM) and RCA detection [63]. The results of the study confirmed the limited geographic distribution of alphasatellites in the New World compared with the Old World, with only two new species detected in whiteflies collected from a single tomato field in Guatemala and Puerto Rico, respectively.

#### Alphasatellite Attenuation of Begomovirus Disease Symptoms May Be Due to α-Rep Interaction with the Helper Virus Rep

The ability of two alphasatellites associated with cotton leaf curl Rajasthan virus (CLCuRaV) to attenuate helper virus levels and symptoms was attributed to the interaction between the α-Rep protein and CLCuRaV Rep, which inhibited virus proper replication [26]. Additionally, asystasia yellow mosaic alphasatellite (AYMA) has been observed to cause yellowing symptoms in *Asystasia gangetica*, a weed common in western and central Africa, during mixed infection with West African Asystasia virus (WAAV), but with no apparent reduction in leaf size [64] (Figure 4). However, the yellowing symptoms are observed to be more intense and systemic in plants containing AYMA, compared with plants singly infected by WAAV. A survey in China showed that plants infected by alphasatellites tended to also contain betasatellites, yet only a few betasatellite-infected plants contained alphasatellites [65]. Thus, alphasatellite modulation of virus levels and disease severity may ensure host survival, which ultimately helps the virus successfully complete its infection cycle. Moreover, alphasatellites may be molecular parasites that reduce the levels of the helper begomovirus, thereby lessening virus disease symptoms.

There have been reports of some Old World alphasatellite-encoded α-Rep increasing the accumulation of helper virus DNA by suppressing RNA silencing, such as with α-Reps of tomato yellow leaf curl China virus (TYLCCNA), the alphasatellites associated with TYLCCNV [66]. Similar results were recorded for the α-Rep of cotton leaf curl Multan alphasatellites (CLCuMuA) associated with CLCuMuV [67,68]. α-Reps of gossypium darwinii symptomless alphasatellite (GDarSLA) and gossypium mustelinium symptomless alphasatellite (GMusSLA) coinfecting *Gossypium* spp. with cotton leaf curl Rajasthan virus (CLCuRaV) were found to be suppressors of RNA silencing, but not βC1 of CLCuMuB, or viral TrAP, C4, and MP, which are known suppressors of RNA silencing [68]. Unsurprisingly, the α-Reps of TYLCCNA, GDarSLA, and GMusSLA that have been shown to suppress RNA silencing group together in a phylogenetic tree (Figure 5). The New World euphorbia yellow mosaic alphabetasatellite (EuYMA)-encoded α-Rep was found to not suppress RNA silencing [69]. Yet, EuYMA, as well as tomato severe rugose alphasatellite (ToSRA), were reported to differentially interact with EuYMV and ToYSV in different hosts. Thus, EuYMA increased symptoms of EuYMV and ToYSV in tomato and *N. benthamiana*, while ToYSA caused more severe symptoms of ToYSV in *N. benthamiana* and *L. sibiricus*, *but negatively affected* the accumulation of ToYSV in both hosts [69,70]. This stresses the roles of helper viruses and host factors in these interactions.

Sequence alignment of these α-Reps showed substantial variation between New World and Old World α-Reps, especially at the N- and C-termini (Figure 3B), which likely explains the dichotomy in these biological functions. Future studies will elucidate the mechanism by which these α-Reps suppress RNA silencing and the basis of the differential behavior between New World and Old World α-Reps. Furthermore, it remains unclear whether α-Rep can trans-replicate the helper virus without the encoded Rep.

### 2.3. Deltasatellites Do Not Enhance Helper Virus DNA Levels or Symptom Severity

The replication of this deltasatellite was shown to be supported by several monopartite and bipartite begomoviruses, including tomato yellow leaf curl virus (TYLCV), African cassava mosaic virus (ACMV), and the curtovirus beet curly top virus (BCTV) [71]. Like Old World deltasatellites, New World deltasatellites do not affect the symptoms induced by the helper begomoviruses and tend to reduce virus accumulation [33]. Viral DNA levels and symptom severity of SPLCV were reported to be lower in the presence of SPLCV-SPLCD1, its associated deltasatellite [72]. However, like alphasatellites, deltasatellites have differential effects on the helper viruses. Recent research indicates that deltasatellites influence the replication dynamics of their helper begomoviruses, either by enhancing or reducing their accumulation [32].

### 2.4. DNA Satellites’ Complementation of Begomoviruses from Different Regions May Lead to New Disease Epidemics

Since begomovirus acquisition or exchange of satellite DNAs may lead to adaptation to new plant hosts, it is important to investigate the diversity and distribution of these molecules. New DNA satellites continue to be identified in association with begomoviruses in different regions, especially in the Indian subcontinent and Africa. Viruses and satellites coevolve together, resulting in a somewhat stable disease dynamic in the host under prevailing environmental conditions. This dynamic may, however, change if new viruses and satellites are introduced from other regions. Thus, complementation between begomoviruses and non-cognate satellites requires investigation. In a recent study, CLCuKoV and CLCuMuV from Pakistan were found to each interact with CLCuGeB from Cameroon synergistically in *N. benthamiana*, which developed more severe symptoms, with higher viral DNA levels compared with plants containing the virus alone [73]. The same study showed that CLCuKoV and CLCuMuV complemented infection by ALCCMB, also from Cameroon, while OYCrV from Cameroon and TYLCV from Australia functionally interacted with CLCuMuB, ALCCMB, and CLCuGeB, respectively. These complementations appear not to be limited to betasatellites, given that the study further showed that all begomoviruses from different geographical origins that were tested supported ALCCMA from Africa [73]. Although these studies were carried out on *N. benthamiana*, an experimental host, there is reason to believe that similar interactions occur in natural hosts. For example, another study showed that CLCuGeB, which is associated with its cognate CLCuGeV in okra, was transmitted and assisted by TYLCV in tomato plants [74]. In the absence of available data, there is a likelihood of high compatibility between alphasatellites and non-cognate helper begomoviruses, given that alphasatellites exhibit autonomous replication. The possibility of functional interactions between satellite and helper virus shows the evolutionary potential of these complexes, which may disturb the established equilibrium and lead to the emergence of new begomovirus epidemics.

## 3. Recombination Between Helper Viruses and DNA Satellites Contributes to Virus and Satellite Evolution

Although the evolutionary relationship between begomoviruses and their DNA satellites, as well as amongst DNA satellites, is still being examined, early evidence suggests the occurrence of recombination events in these molecules’ evolution [75]. This is expected, given that helper viruses and their satellites replicate using recombination-dependent replication. Thus, it is unsurprising that recombination was reported between ageratum yellow vein virus (AYVV) and its associated betasatellite (AYVB) and with the alphasatellite (AYVA) in coinfected *Ageratum conyzoides* plants [23,31,76]. The recent identification of a chimeric DNA molecule with a similar size to alphasatellites and betasatellites and consisting of sequences derived from CLCuKoV, CLCuMuA, and croton yellow vein mosaic alphasatellite [77] further supports ongoing recombination between helper virus and associated DNA satellites. Additionally, recombination events have been reported between CLCuKoV and CLCuMuV with their respective associated satellites, CLCuMuB and CLCuMuA [78,79,80]. Moreover, in Pakistan, a recombinant DNA betasatellite with a documented epidemiological impact on cotton plantings was found to contain a sequence derived from CLCuMuB, as well as from tomato leaf curl betasatellite [81], causing the breakdown of resistance to CLCuKoV [57,82,83,84].

Recombination events are not limited to protein-coding alphasatellites and betasatellites. Half of the genome of SPLCV-SPLCD1, the deltasatellite associated with SPLCV, is thought to have been acquired from the latter as a result of a recombination event in coinfected *Convolvulaceae* and/or sweet potato [77] (Figure 2C). The role of recombination in interactions between helper viruses and DNA satellites is evidenced by many new begomovirus epidemics involving recombinant molecules. This is evident, particularly in the Indian sub-continent, where there is currently more documentation of begomovirus–DNA satellite complexes [44,50,57,75,77,78,79,80,82,83].

## 4. Cassava Geminivirus-Associated SEGS-1 and SEGS-2 Episomes Are Novel DNA Satellites

Two episomes designated as SEGS-1 and SEGS-2—sequences enhancing geminivirus symptoms—isolated in cassava plants exhibiting unusually severe CMD symptoms have been reported in cassava fields in the coastal region of Tanzania and the tropical forest region of Cameroon [85,86]. In a comprehensive survey carried out in Ghana, West Africa, out of 110 cassava leaf samples affected by cassava mosaic begomoviruses, SEGS-2 and SEGS-1 were detected in 66 (60%) and 47 (43%), respectively [87]. It remains unclear whether this high incidence is representative of the region. The two episomes (Figure 6) share no sequence identity with each other, the associated cassava geminiviruses, or the three DNA satellites discussed in this review. SEGS-1 shows a 98% sequence identity with a satellite DNA molecule isolated in mentha (*Mentha spicata*) plants infected by tomato leaf curl Karnataka virus-Bangalore (ToLCKV-Ban) in Punjab, India [88]. This molecule was designated as *mentha leaf deformity*-associated DNA-II (MLDA-DNA-II) because of the deformation it caused in association with ToLCKV-Ban in coinfected mentha plants. Thus, SEGS-1, which, like SEGS-2, induces deformed cassava leaves and enhanced CMD symptoms in cassava (*Manihot esculenta* Crantz) in coinfection with cassava mosaic geminiviruses (CMGs) [86], causes similar symptoms in mint plants in India.

Sequences related to SEGS-1 and SEGS-2 occur in the cassava genome. For example, a third of SEGS-1 is 99% identical to a segment of the eukaryotic ORF encoding the protein prefoldin subunit 6 (PFDN6), which is a subunit of the heteromeric prefoldin complex involved in the proper folding of cytoskeletal proteins [89,90]. Correspondingly, over a third of SEGS-2 shows 96% sequence identity with the cassava pentatricopeptide repeat (PPR) family of proteins, which regulate gene expression in chloroplasts and mitochondria and are involved in many aspects of plant growth and development, including responses to stress and reproduction [91]. Additionally, the 3’ half of PPR in SEGS-2 contains a start codon of a potential 68-amino acid peptide. This sequence maps to the first 68 amino acids of pre-mRNA-splicing factor ATP-dependent RNA helicase DEAH1 isoform X2 (DHX16 isoform X2). This DHX16 isoform X2 is part of the spliceosome complex involved in the processing of pre-mRNA into mRNA [92]. DHX16 isoform X2 and PPR are found in the same region of the cassava genome (loci XM_043949572 and XM_043949652), and their ORFs overlap but are in different frames. Sequence alignment of SEGS-2 and these two sequences shows the specific DNA fragment that became part of the SEGS-2 molecule (Figure 7). Furthermore, another segment of the SEGS-2 genome shows 83% sequence identity with a noncoding region of the cassava genome but outside the DHX16 isoform X2 and PPR loci. This adds complexity regarding the origins and evolutionary path of SEGS-2.

Other structural features of these episomes include a GC-rich region that covers a third of each molecule, with a CpG/GpC ratio of 0.84, therefore qualifying as CpG islands [93]. These CpG islands are positionally located in the recombinant PFDN6 and PPR regions, respectively (Figure 6). CpG islands are substrates for DNA methylation, especially in promoter regions, leading to gene silencing [94,95,96,97]. The role played by a heavily methylated region in these episomes’ biology is yet to be elucidated. SEGS-1 episomes are only found in CMG-infected plants, consistent with infection-promoting episome formation [98]. Given that begomovirus infection induces the host cell cycle S phase and DNA replication [99,100] and interferes with DNA methylation [101], the cell replicative state and/or the decrease in DNA methylation could plausibly lead to the activation and generation of SEGS-1 episomes [98]. This supports the view that, unlike SEGS-2, SEGS-1 is not a DNA satellite. Furthermore, SEGS-1 is ubiquitous in the cassava genome; thus, it was suggested to be perceived as a member of a repetitive sequence family that is inactivated by DNA methylation and/or sequestration into heterochromatin [102].

Both episomes display characteristic stem–loop structures. With a confidence level of >90%, this feature is real and likely has an important biological role, and it was suggested to harbor the replication origin of the episomes [86]. Future investigations will determine whether these hairpins are packaging signals for SEGS-2, which is packaged by the helper CMGs [86].

Epidemiologically, SEGS-1 and SEGS-2 enhance CMG replication and symptom severity and were reported to cause a break in resistance to CMD in cassava in the coastal region of Tanzania [85,86,102]. Correspondingly, in Cameroon, elite CMD-resistant cassava breeding lines containing the polygenic recessive resistance locus and a dominant monogenic resistance locus displayed severe symptoms when coinfected by these episomes and ACMV. In more severe cases, the leaf is reduced to the midrib, with a total absence of the leaf blade (Figure 8). These episomes were also found to cause ACMV infection and the enhancement of symptoms when applied exogenously to Arabidopsis leaves, as well as in Arabidopsis containing an integrated copy of the episome [85,102]. This indicates these episomes’ ability to expand the host range of ACMV. The SEGS-2 encoded putative 68-amino acid protein that corresponds to DHX16 isoform X2 protein likely plays a role in severe CMD symptoms observed in coinfected plants [85]. Because SEGS-2 is packaged into virions and transmitted by whiteflies along the helper CMG [85,86], this episome qualifies as a DNA satellite.

A recent in-depth study has shown that, unlike SEGS-2, SEGS-1 activity in Arabidopsis is not mediated by an episome, but rather by the transgene and is, therefore, chromosomal [102]. This is supported by SEGS-1 molecules having only been found in plants and not in virions or whiteflies, suggesting that they may originate from the cassava genome. Thus, SEGS-1 is likely not a DNA satellite. This is supported by infected plants of some, but not all, cassava cultivars forming SEGS-1 episomes during CMG infection in the absence of exogenous SEGS-1 [102].

## 5. Outlook

Considerable efforts are required to identify all components underpinning the global begomovirus epidemics to contain these epidemics. This will enable us to design appropriate control strategies. Here, some immediate research efforts to understand the epidemics, as well as the use of prime editing to generate potentially non-GMO virus-resistant plants, are discussed.

### 5.1. Knowledge Gaps That Need Addressing on Begomovirus Diseases

Future research efforts will need to determine whether DNA satellites influence the efficiency of whitefly transmission of the helper virus. Furthermore, an in-depth comparison between viruliferous whitefly and virus-free whitefly transmission efficiency will provide additional information on the development of begomovirus epidemics. Moreover, it is unclear whether higher virus levels in infected hosts necessarily translate to a higher transmission efficiency. The role of Rep binding in betasatellites in the interaction between the latter and helper begomoviruses needs to be elucidated, given that betasatellites lack the iteron sequences of their helper begomoviruses. Thus, different Reps likely bind with different efficiencies, thereby influencing replication. For example, ToLCKuV from Togo and ToLCNGV, both of which coinfect different host species in association with ToLCTGB, causing different symptoms, have different iterons, which likely influences ToLCTGB’s binding and, thus, infectivity [48]. Thus, investigating the role of the characteristic high affinity of Rep-binding sites in New World begomoviruses, as opposed to betasatellite-binding sites, would address a possible role of Rep-binding sites in the absence of betasatellites in the New World. Furthermore, additional investigation would determine whether α-Reps can efficiently trans-replicate the parent virus, given that such a function would add to our understanding of virus infection.

Furthermore, tomato leaf curl New Delhi virus (ToLCNDV), a causal agent of ToLCD, and begomoviruses responsible for TYLCD are extending the host range and spreading to new geographical regions in the Middle East and the western Mediterranean Basin [103,104,105]. The role of DNA satellites in this expansion will need to be determined in future research efforts. Correspondingly, more comprehensive studies on the diversity, distribution, and impact of DNA satellites are needed in West Africa, where continuing studies are documenting the occurrence of new begomoviruses [46,47,48,51,64,87,106,107,108].

The status of SEGS-1 must also be further addressed. Although it was recently shown not to be transmitted by the whitefly, and, therefore, does not qualify as a DNA satellite [98], this episome will need to be monitored to see if it becomes a DNA satellite. This will provide important information on the origins and evolution of these DNA satellites.

### 5.2. DNA Satellites and Biotechnological Management of Begomovirus Diseases

RNA satellites that attenuate symptoms of the helper RNA viruses have been shown to modulate symptoms and/or levels of the viral RNA, including RNA satellites associated with cucumber mosaic virus [109,110,111], tobacco ringspot virus [112], and bamboo mosaic virus [113]. Reduced virus levels are attributed mainly to competition between satellite RNA and viral genomic RNA for replication [110,114,115]. Bamboo mosaic virus RNA satellites have been used to generate virus-resistant plants [113]. This approach has not yet been used in geminivirus control. The fact that alphasatellites and deltasatellites modulate the helper virus symptoms suggests a natural defense mechanism and, therefore, can be harnessed to produce begomovirus-resistant plants. With recent advances in biotechnological strategies in plant virus management [116,117,118,119], the use of satellite DNA molecules in controlling begomoviruses needs to be explored. For example, prime editing (PE), the novel CRISPR-based genome-editing method, can be used for targeted and precise genome modification [120]. A potentially impactful strategy is to explore the phased secondary small interfering RNA (phasiRNA) pathway. Along with others, we have reported datasets of small RNAs of several crops impacted by begomoviruses and associated satellites [121,122,123,124,125]. These studies include phasiRNAs and their PHAS loci from which 21–24 nt phasiRNA are processed under natural conditions. This provides an excellent opportunity to employ PE in precisely inserting 21–24 mers derived from viruses and/or satellites to generate virus-resistant plants [119]. Thus, the specific sequences of alphasatellites and deltasatellites that enable these DNA satellites to modulate virus symptoms will need to be identified and then inserted into the plant genome using PE.

## Figures and Tables

**Figure 1 ijms-26-05814-f001:**
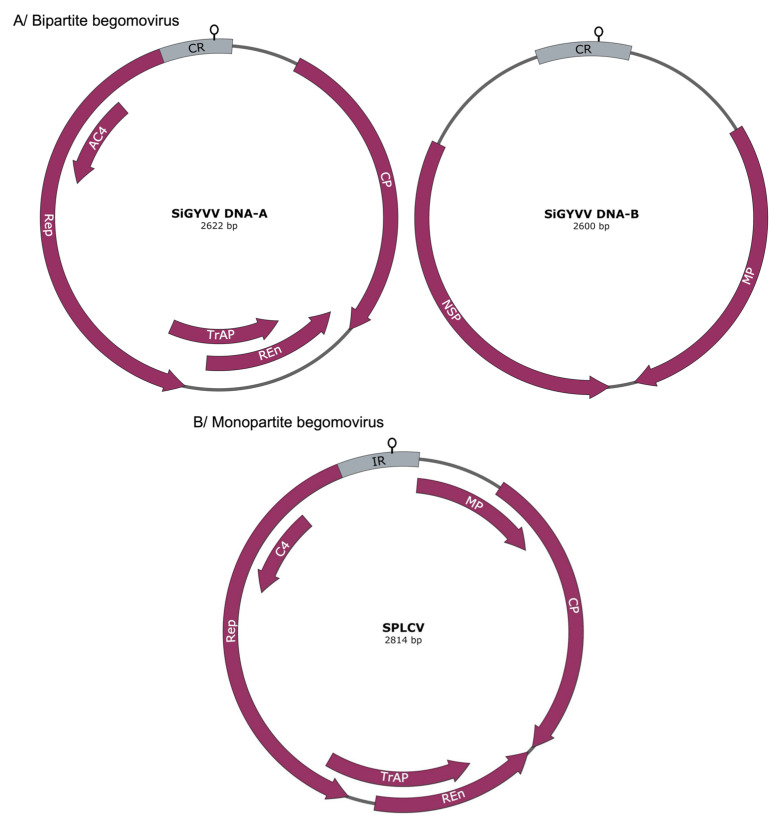
Genome organization of bipartite and monopartite begomovirus genomes. (**A**) Bipartite begomovirus sida golden yellow vein virus (SiGYVV) DNA-A and DNA-B genome components and (**B**) monopartite begomovirus sweet potato leaf curl virus (SPLCV). Canonical viral proteins are represented by curved arrows and include coat protein (CP), replication-associated proteins (Reps), transcriptional activator protein (TrAP), replication enhancer protein (REn), C4, movement protein (MP), and nuclear shuttle protein (NSP).

**Figure 2 ijms-26-05814-f002:**
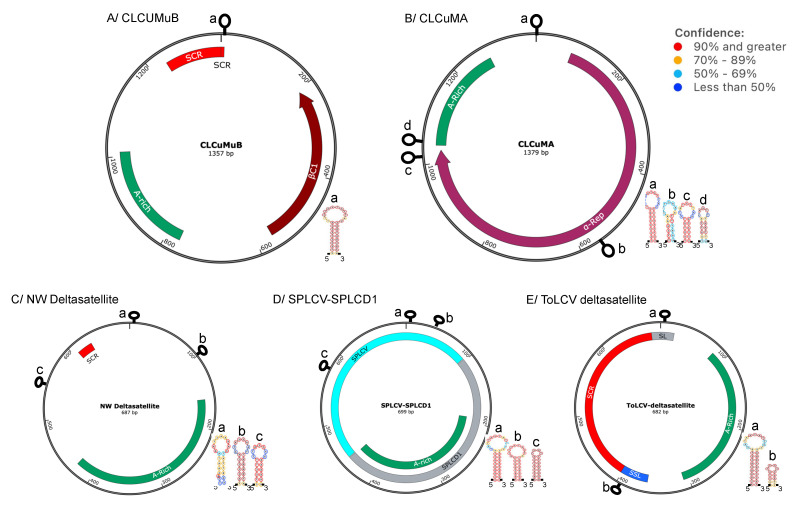
Representative genomes of begomovirus DNA satellites discussed in this review. (**A**) Cotton leaf curl Multan betasatellite (CLCuMuB), (**B**) cotton leaf curl Multan alphasatellite (CLCuMuA), (**C**) New World deltasatellite (NW Deltasatellite), (**D**) sweet potato leaf curl deltasatellite (SPLCV-SPLCD1), and (**E**) tomato leaf curl deltasatellite (ToLCD). Betasatellite encodes βC1, which is a pathogenicity determinant, while alphasatellites encode the α-Rep protein for autonomous DNA replication. The hairpin structures shown harbor the DNA replication origin (a), and CLCuMuA, ToLCD, and NW deltasatellites show other predicted hairpins identified as (b), (c), and (d). The confidence levels of these structures were predicted using the SnapGene software 8.1 (GLS Biotech, San Diego, CA, USA).

**Figure 3 ijms-26-05814-f003:**
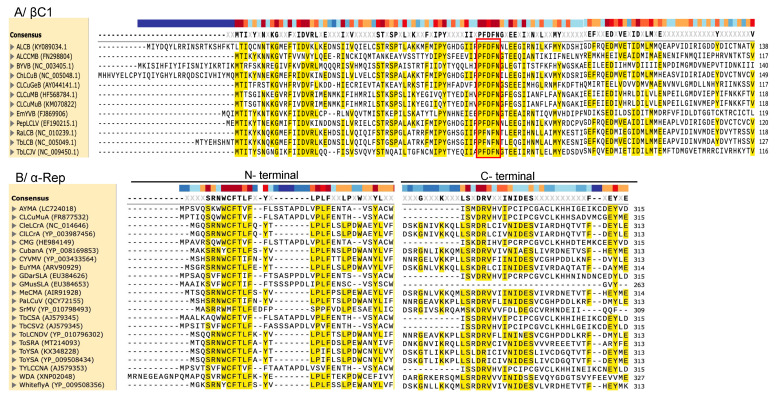
Proteins encoded by betasatellites and alphasatellites show variation. (**A**) βC1 sequence variation throughout the protein sequence, except for the PFDFN motif (red box); (**B**) there is considerable sequence variation between New World and Old World α-Reps, especially at the N- and C-termini. Conserved amino acid residues are highlighted yellow.

**Figure 4 ijms-26-05814-f004:**
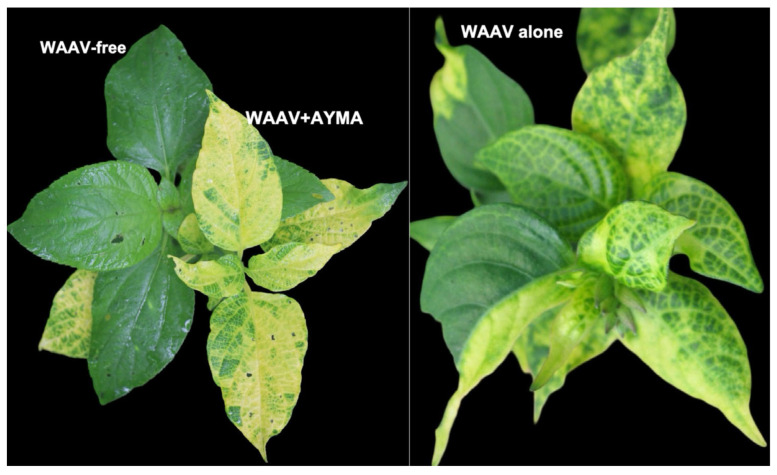
Asystasia yellow mosaic alphasatellite (AYMA) causes intense yellowing symptoms in *Asystasia gangetica* in the presence of West African asystasia virus (WAAV).

**Figure 5 ijms-26-05814-f005:**
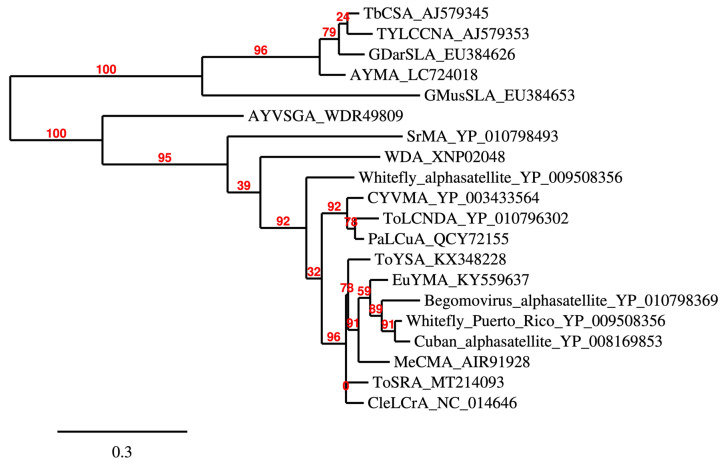
Phylogenetic tree generated using maximum-likelihood method from alignment of α-Reps. α-Reps of GDarSLA and GMusSLA that suppress PTGS group together in a phylogenetic tree.

**Figure 6 ijms-26-05814-f006:**
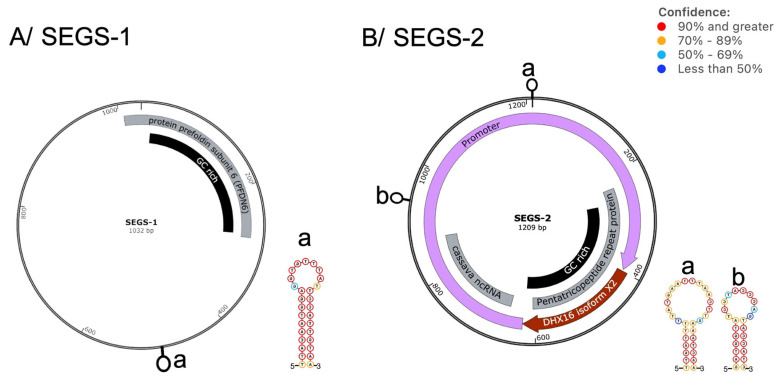
Representation of cassava mosaic geminivirus episomes. (**A**) SEGS-1 sequence has identity with the cassava protein prefoldin subunit 6 (PFDN6). (**B**) SEGS-2 shows sequence identity with the cassava pentatricopeptide repeat (PPR) family, as well as a potential protein of 68 amino acids that maps to the N-terminal first 68-amino acid segment of the pre-mRNA-splicing factor ATP-dependent RNA helicase DEAH1 isoform X2 (DHX16 isoform X2). Like SEGS-1, SEGS-2 has a CpG island, as well as two predicted stem–loop structures. Episome stem-loops are represented by a and b. The confidence levels of these structures were predicted using the SnapGene software 8.1 (GLS Biotech, San Diego, CA, USA).

**Figure 7 ijms-26-05814-f007:**
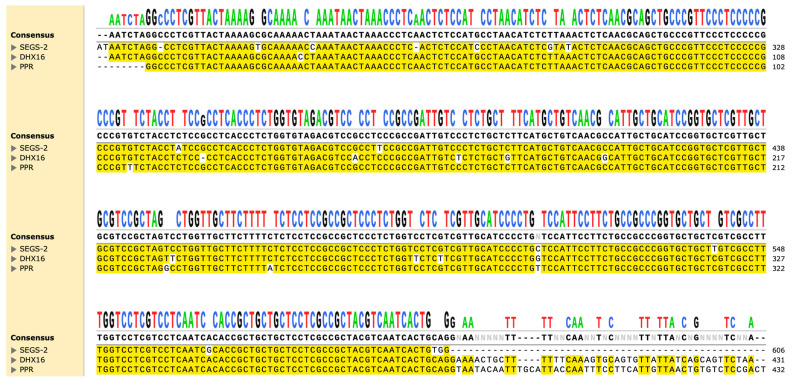
Sequence alignment of SEGS-2, PPR; DHX16 isoform X2 shows specific DNA fragment that became part of SEGS-2 molecule. Conserved nucleotides are highlighted yellow.

**Figure 8 ijms-26-05814-f008:**
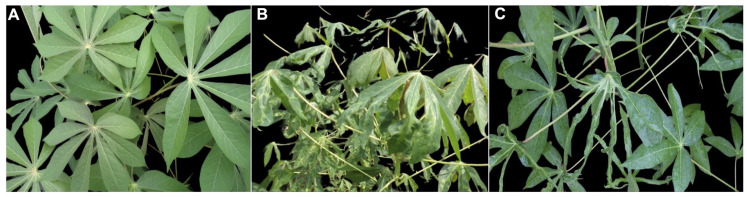
SEGS episomes cause severe symptoms when mixed-infected with cassava mosaic geminiviruses. CMG-resistant cassava showing (**A**) ACMV-free plant, (**B**) plant infected by ACMV alone, and (**C**) plant mixed-infected by episomes and ACMV displays leaves that are reduced to the midrib with a total absence of the leaf blade.

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
