# Peer review of "DNA Satellites Impact Begomovirus Diseases in a Virus-Specific Manner"

_ijms, 2025, doi:10.3390/ijms26125814_

Round 1

Reviewer 1 Report

Comments and Suggestions for Authors

This paper is well organized and well written, comprehensively presenting the association between DNA Satellites and Begomovirus Diseases. I only have two questions: 1)Whether and how these DNA satelites impact immune responses in plants; 2) future developments, such as drugs (small peptides, molecules) or preventions against the virus by exploiting DNA satelites could be added and discussed in depth.

Author Response

Reviewer 1

This paper is well organized and well written, comprehensively presenting the association between DNA Satellites and Begomovirus Diseases. I only have two questions: 1) Whether and how these DNA satellites impact immune responses in plants

Response: I have included in the text the fact that of the three satellites developed, betasatellites enhance virus symptoms and thus weaken the host immune system, in contrast, alphasatellites and deltasatellites tend to moculate symptoms.

2) future developments, such as drugs (small peptides, molecules) or preventions against the virus by exploiting DNA satellites could be added and discussed in depth.

Response: A potentially impactful approach to exploit DNA satellites to enhance plant resistance is the use of PHAS RNA pathway. More details on this approach have been included.

These changes have been highlighted in the text.

Reviewer 2 Report

Comments and Suggestions for Authors

I have included some comments within the text for the author's consideration. Regarding organization, I suggest repositioning Figure 4 to a location after line 291, where it is first cited in the text.

Author Response

All the edits and comments suggested by the reviewer have been made. These revisions are highlighted green in the text. Figure 4 has been moved as requested.